# Lecithin-Based Dermal Drug Delivery for Anti-Pigmentation Maize Ceramide

**DOI:** 10.3390/molecules25071595

**Published:** 2020-03-31

**Authors:** Kazuhiro Kagotani, Hiroko Nakayama, Liqing Zang, Yuki Fujimoto, Akihito Hayashi, Ryoji Sono, Norihiro Nishimura, Yasuhito Shimada

**Affiliations:** 1Tsuji Health & Beauty Science Laboratory, Mie University, Tsu 514-8507, Japan; k.kagotani@tsuji-seiyu.co.jp; 2Zebrafish Drug Screening Center, Mie University, Mie 514-8507, Japan; 27293301@m.mie-u.ac.jp (H.N.); liqing@doc.medic.mie-u.ac.jp (L.Z.); nishimura.norihiro@mie-u.ac.jp (N.N.); 3Graduate School of Regional Innovation Studies, Mie University, Tsu 514-8507, Mie, Japan; 4Tsuji Oil Mills Co., Ltd., Matsusaka, Mie 515-0053, Japan; y.fujimoto@tsuji-seiyu.co.jp (Y.F.); a.hayashi@tsuji-seiyu.co.jp (A.H.); r.sono@tsuji-seiyu.co.jp (R.S.); 5Department of Bioinformatics, Mie University Advanced Science Research Promotion Center, Tsu 514-8507, Mie, Japan; 6Department of Integrative Pharmacology, Mie University Graduate School of Medicine, Tsu 514-8507, Mie, Japan

**Keywords:** zebrafish screening, cone ceramide, in vivo imaging

## Abstract

Ceramides have several well-known biological properties, including anti-pigmentation and anti-melanogenesis, which make them applicable for use in skincare products in cosmetics. However, the efficacy of ceramides is still limited. Dermal or transdermal drug delivery systems can enhance the anti-pigmentation properties of ceramides, although there is currently no systemic evaluation method for the efficacy of these systems. Here we prepared several types of lecithin-based emulsion of maize-derived glucosylceramide, determining PC70-ceramide (phosphatidylcholine-base) to be the safest and most effective anti-pigmentation agent using zebrafish larvae. We also demonstrated the efficacy of PC70 as a drug delivery system by showing that PC70-Nile Red (red fluorescence) promoted Nile Red accumulation in the larval bodies. In addition, PC70-ceramide suppressed melanin in mouse B16 melanoma cells compared to ceramide alone. In conclusion, we developed a lecithin-based dermal delivery method for ceramide using zebrafish larvae with implications for human clinical use.

## 1. Introduction

Ceramides have a variety of biological functions as second messengers in several cellular events, including proliferation, differentiation, cell cycle arrest, apoptosis, and senescence [1,2,3,4], in addition to the role of ceramides as moisturizers used in cosmetic products. Since Kim et al. (2001) demonstrated that C2-ceramide (N-Acetyl-D-erythro-sphingosine), a cell-permeable ceramide analogue, suppressed melanoma cell proliferation while reducing melanin synthesis [5], several studies suggested a possible use for ceramides as anti-pigmentation agents [6,7,8]. However, ceramides are of limited efficacy regarding skin pigmentation reduction because they are poorly soluble in water. We thus hypothesized that dermal or transdermal drug delivery for ceramide would enhance the anti-pigmentation properties of these molecules.

In accordance with “the three Rs (Replacement, Refinement, and Reduction)” principle consistent with standard ethics committee guidelines, zebrafish were proposed as a model organism for pigmentation studies in the medicinal and cosmetic research fields. Recent studies demonstrated a close correlation between zebrafish and human pigmentation [9,10,11], with many depigmenting agents identified using zebrafish embryos or larvae [12,13,14,15]. In addition, zebrafish are now used to test several types of drug delivery systems, including liposomes [16,17], micelles [18], and nanoparticles [19,20,21,22].

Lecithin is mixtures of glycerophospholipids, including phosphatidylcholine, phosphatidylethanolamine, phosphatidylinositol, phosphatidylserine, and phosphatidic acid, which are essential in animal and plant cells. Glycerophospholipids, such as phosphatidylcholine, exhibit amphiphilic structures [23], and can be used as inactive ingredients to form novel mixtures for smoothing food textures, emulsifying, homogenizing liquid mixtures, and repelling sticking materials. Indeed, lecithin has been applied extensively in nanoscale drug delivery systems in recent years [24,25,26].

In this study, we prepared three types of lecithin emulsions of ceramide (maize-derived glucosylceramide) and exposed them to zebrafish larvae to evaluate their efficacy as depigmentation agents. We identified a recipe for the emulsion to enhance the depigmentation effect of ceramide while also minimizing toxicity. We likewise created a lecithin emulsion of Nile Red, a red fluorescent dye, to visualize drug delivery in vivo.

## 2. Results

### 2.1. PC70-Ceramide Suppresses Pigmentation in Zebrafish

We used three types of lecithin containing phosphatidylcholine (PC), lysophosphatidylcholine (LPC), and hydrogenated lysophosphatidylcholine (LPCH) to emulsify maize-derived ceramide, denoted PC70, LPC70, and LPC70H (Figure 1a). These lecithin-ceramides contained 20 mg/mL of each lecithin and 10 mg/mL ceramide. We first tested the biological safety of lecithin-ceramide emulsions and ceramide alone up to concentrations of 500 µg/mL during zebrafish development from 24 h post-fertilization (hpf). As shown in Figure 1b, the lecithin-ceramides showed a variety of toxicities on 6 d post-fertilization (dpf) larvae (5 d exposure); the structures of their lecithin components were also similar (Figure 1a). The LC_50_ values for PC70-, LPC70- and LPC70H-ceramides were measured in terms of ceramide concentrations as 266.5 ± 101.8 µg/mL, 1.7 ± 0.2 µg/mL, and 1.1 ± 0.5 µg/mL, respectively. Ceramide alone exhibited no toxicity up to 500 µg/mL. The corresponding LC_50_ values of lecithin alone (without ceramide) are indicated in Table 1 (represented values are lecithin concentrations).

We then exposed these lecithin-ceramide emulsions to dechorionated 24 hpf embryos for 48 h. We found that 0.1% PC70-ceramide, which contained 20 µg/mL PC and 10 µg/mL ceramide, significantly (*p* < 0.05) reduced pigmentation in the larvae compared to the control and ceramide-alone treatments (Figure 1c,d). Other lecithin-ceramides (LPC70- and PC70H-ceramides) did not show any effects on pigmentation at 1 and 0.5 µg/mL ceramide concentrations (about half of their LC_50_ concentrations), respectively (data not shown).

### 2.2. Nile Red Delivery by PC70-Lecithin

To visualize the PC70-induced dermal ceramide delivery in zebrafish, we prepared PC70-Nile Red (NR) emulsions instead of PC70-ceramide. Two hours after PC70-NR exposure, 1% PC70-NR groups (containing 200 μg/mL PC70 with 1 ng/mL NR) showed higher red fluorescence than those treated with NR alone (Figure 2a, upper panels); this pattern continued until 20 h exposure (Figure 2a, middle panels). Interestingly, 18 h after washing the zebrafish exposed for 20 h, PC70-NR groups showed higher fluorescent signal than NR groups (Figure 2a, lower panels), suggesting that PC70 promotes dermal ceramide delivery and retention inside the skin. Quantitative analysis of these NR signals is presented in Figure 2b.

### 2.3. Lecithin-Ceramide Treatment in Mouse Melanoma Cells

The mouse B16 melanoma 4A5 cell line is known to be a suitable in vitro model for melanin synthesis [27]. We used this cell line to test whether the lecithin-ceramides used in this study could suppress melanogenesis in vitro. We found that 0.5% PC70-ceramide (equal to 50 µg/mL ceramide) significantly (*p* < 0.05) reduced melanin accumulation in B16 cells, while 50 µg/mL ceramide alone did not reduce melanin content (Figure 3a,b). In addition, 100 µM arbutin, which is known to inhibit tyrosinase activity [28], did not significantly reduce melanin content (*p* = 0.2), in contrast to the results obtained in previous studies [8]. The lecithin-ceramide emulsions demonstrated no apparent toxicity during the experiment (Figure 3c).

## 3. Discussion

### 3.1. PC70-Lecithin is the Safest Emulsifier for Dermal Drug Delivery in Zebrafish

In this study, we tested three types of lecithin-ceramides, phosphatidylcholine (PC70), lysophosphatidylcholine (LPC70), and hydrogenated lysophosphatidylcholine (LPC70H), and found that PC70 was the safest lecithin emulsion for zebrafish larvae (Figure 1b). The LC_50_ of PC70 was around 100 times lower than that of LPC70 and LPC70H, indicating that lysophosphatidylcholine (LPC) is more toxic than phosphatidylcholine (PC). Previous studies reported that LPC has hemolytic properties [29] and causes damage to cardiac cells [30], although LPC has also been shown to be able to penetrate human skin cells without damaging the skin structure [31]. We thus assumed that LPC exhibited high toxicity toward zebrafish larvae in this study due to the body wall of zebrafish larvae being more permeable than human skin. Interestingly, all of the lecithin used in this study (PC70, LPC70, and LPC70H) showed lower LC_50_ values (higher toxicity) compared to lecithin-ceramide (Table 1). Ceramide alone did not show toxicity at a concentration of 500 µg/mL, even in its insoluble form, indicating that the structures of lecithin-ceramide emulsions reduce lecithin toxicity. We considered this to be due to the fact that when PC70 is encapsulated by a hydrophobic component such as ceramide, the hydrophobic substrates are oriented facing each other, therefore the structure of the emulsion is stable. When PC70 is not encapsulated, the stability of the structure is poor, and hydrolysis of PC70 occurs when dissolved in water. This leads to the conversion of PC70 to LPC70, a highly toxic lecithin.

### 3.2. PC70-Based Ceramide Delivery

While ceramide and its bioactive metabolites help maintain skin homeostasis, including depigmentation, ceramide itself is too hydrophobic to be dissolved in the skin liniment base, therefore it exhibits relatively low efficacy in vivo. Indeed, during our ceramide exposure experiments in zebrafish, concentrations higher than 5 µg/mL ceramide showed insoluble precipitation, thereby reducing their biological efficacy (Figure 1c,d). Since lecithin is amphiphilic compounds, PC70-ceramide was completely dissolved at concentrations up to 250 µg/mL ceramide, thus demonstrating solubility 50 times higher than that of ceramide alone and proving a great advantage to using lecithin emulsions as drug delivery systems. In addition, PC70-ceramide containing 50 µg/mL ceramide suppressed melanin accumulation in both zebrafish larvae and cultured melanocytes. Besides its use in cosmetics, ceramide and its bioactive metabolites are involved in epidermal self-renewal and immune regulation, which provides new opportunities for therapeutic intervention in inflammatory dermatoses, such as psoriasis and atopic dermatitis [32,33].

## 4. Materials and Methods

### 4.1. Preparation of Lecithin Emulsion

The three types of lecithin used in this study (PC70, LPC70, and LPC70H (Tsuji Oil Mills, Mie, Japan)) are described in Figure 1a. The epithet PC70 indicates that phosphatidylcholine accounts for more than 70% of the total lecithin. Two percent (w/w) of each lecithin (equal to 20 mg/mL) was dissolved in 70% glycerol at 60 °C. A further 1% (w/w) glucosylceramide from maize (equal to 10 mg/mL; Wako Pure Chemicals, Osaka, Japan) or oil containing 10 ppm Nile Red (NR; Wako Pure Chemicals) was then added to the mixture; the mixture completely dissolved at 60 °C. Preliminary emulsification of the mixture was performed at 60 °C for 5 min, using a disperser (Polytron Homogenizer PT2100, Central Scientific Commerce, Tokyo, Japan). The emulsion was then treated twice at 100 MPa using the NM2-L200 emulsifying machine (Yoshida Kikai, Aichi, Japan). The resulting emulsions used in this study contained 2% of each lecithin with 1% ceramide or 100 ng/mL NR. While penetration ability of dimethyl sulfoxide (DMSO) in skin tissues is better than that of glycerol [34], we chose glycerol because it maintains barrier functions of the skin [35]. The average emulsified particle diameters of PC70-ceramide and PC70-NR were 0.070 μm and 0.794 μm, respectively. These were analyzed using a particle size distribution analyzer (LS 13 320, Beckman Coulter, CA, USA). PC70-ceramide was imaged using a TM-1000 scanning electron microscope (Hitachi, Tokyo, Japan).

### 4.2. Ethics

All animal procedures were performed according to the Japanese animal welfare regulatory practice Act on Welfare and Management of Animals (Ministry of Environment of Japan), in compliance with international guidelines. Ethical approval from the local Institutional Animal Care and Use Committee was not sought, as this law does not mandate the protection of fish.

### 4.3. Zebrafish Experiment

The zebrafish wild AB strain was purchased from the Zebrafish International Resource Center (Eugene, OR, USA) and maintained in our facility according to standard operational guidelines. Twenty-four hours post-fertilization (hpf) embryos were dechorionated using 2 mg/mL Pronase solution (Sigma-Aldrich, St Louis, CA, USA), following the methodology used in previous studies [36]. Following this, 12 embryos were transferred in 4 mL of 0.3× Danieau’s solution (fish medium) to six-well plates with or without lecithin-ceramides, and incubated for 96 h at 28 °C. At 72 hpf, images of the larvae were captured using a BZ-X710 microscope (Keyence, Tokyo, Japan). To evaluate the toxicity of the tested compounds, the larvae were exposed to the emulsions for five days. The fish medium was refreshed every other day during the experiment. The number of survivals was recorded daily. For the PC70-NR experiment, we added 1% PC70-NR or 1 ng/mL NR solution to 72 hpf zebrafish larvae and incubated these for 2 and 20 h, respectively, at 28 °C. The fish were then washed with breeding water for 18 h. Fluorescent images of the larvae were captured using a BZ-X710 microscope (Keyence).

### 4.4. Measurement of Pigmentation in Zebrafish Larvae

We used ImageJ (Fiji distribution, version 1.52p, National Institute of Health, Bethesda, MD, USA) to measure the degree of pigmentation in the zebrafish larvae. The images were first converted to 8-bit grey images and the threshold was set to select only the pigmented area. To remove artefacts (eyes and shadows around the yolk sac) from the images, we set the “analyze particles” parameter with an appropriate pixel size limit. For the Nile Red experiment, the integrated density of each image was measured as Nile Red fluorescent intensity using ImageJ, following the methodology used in a previous study [37].

### 4.5. Mouse Melanoma Cells

Mouse B16 melanoma 4A5 cells [38] were obtained from RIKEN Cell Bank (Ibaraki, Japan) and were cultured in α-minimum essential medium (Nacalai Tesque, Tokyo, Japan) with 10% fetal bovine serum (Biowest SAS, Nuaille, France) and 1% penicillin–streptomycin solution (Wako Pure Chemicals, Osaka, Japan) at 37 °C in 5% CO_2_.

### 4.6. In Vitro Pigmentation Assay

Melanin content in the mouse cells was measured following the methodology of a previous study [39], with slight modifications. Mouse melanoma B16 4A5 cells were seeded in a six-well plate (6.7 × 10^3^ cells/cm^2^) and cultured for 24 h in the complete medium. The cells were then transferred to the medium containing 1 mM theophylline (Wako Pure Chemical) and treated with each concentration of either ceramide alone or lecithin-ceramides for 48 h. The cells were collected using trypsin treatment and centrifugation (200× *g* for 5 min), suspended in 0.5 mL of the culture medium, and counted using a TC20 cell counter (Bio-Rad Laboratories, Hercules, CA, USA). The cells were then stained with 0.4% trypan blue (Bio-Rad Laboratories) to count the number of viable cells. The cell suspensions were then centrifuged again. The cell pellets were then lysed using 10% DMSO/2N NaOH solution (200 μL each) at 60 °C for 30 min. Samples containing 150 μL of lysis solution were transferred into 96-well plates, and the absorbance of each sample was measured at 405 nm/655 nm using an iMark microplate reader (Bio-Rad Laboratories). The data were then converted into absorbance per cell number using a cell number of 2.0 × 10^6^ cells.

### 4.7. Statistics

The data were analyzed statistically using Student’s t-test or one-way analysis of variance according to the Bonferroni–Dunn multiple comparison procedure, depending on the number of comparisons to be performed, using GraphPad Prism version 8 (GraphPad Software, San Diego, CA, USA). The LC_50_ values were calculated with GraphPad Prism version 8 software. A *p*-value less than 0.05 denoted the presence of a statistically significant difference between treatments.

## 5. Conclusions

We developed a lecithin-based ceramide delivery method for zebrafish larvae, which successfully suppressed pigmentation. PC70 could be further applied to emulsify other hydrophobic compounds to increase their biological efficacy in vivo. Hopefully, our lecithin-based delivery system could be applied not only to dermal delivery of ceramides but also to other types of administration, such as intravenous or intraperitoneal.

## Figures and Tables

**Figure 1 molecules-25-01595-f001:**
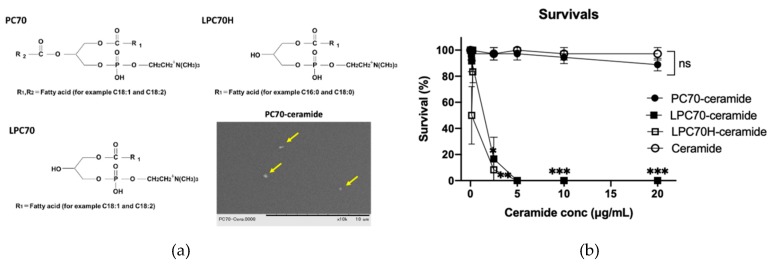
PC70-ceramide suppresses pigmentation in zebrafish. (**a**) Structures of lecithin used in this study and electron microscope image of PC70-ceramide (10,000× magnification; yellow arrows). (**b**) Survival rate for zebrafish larvae treated with lecithin-ceramides. The fish were treated with each compound from 24 h post-fertilization (hpf) for 5 days. * *p* < 0.05, ** *p* < 0.01, and *** *p* < 0.001 vs. control. n = 3, error bars indicate SD. ns indicates no significant difference. For the survival rate for lecithin alone, please see Table 1. (**c**) Representative images of zebrafish larvae treated with 0.1% PC70-ceramide (containing 10 µg/mL ceramide), 0.1% PC70, or 10 µg/mL ceramide. The fish were treated with each compound from 24 hpf for 48 h. (**d**) Quantification of pigmentation in zebrafish larvae treated with PC70-ceramide, PC70, or ceramide. * *p* < 0.05 vs. control. n = 12, error bars indicate SD.

**Figure 2 molecules-25-01595-f002:**
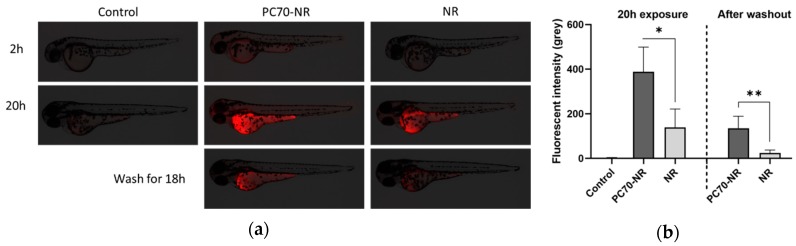
Visualization of the PC70 drug delivery system using PC70-Nile Red (PC70-NR) treatment in zebrafish larvae. (**a**) Representative images of PC70-NR- or NR-treated zebrafish. Seventy-two hpf larvae were exposed to 1% PC70-NR (containing 200 μg/mL PC70 with 1 ng/mL NR) or 1 ng/mL NR for 2 and 20 h, respectively. Following this, the fish were washed in breeding water for 18 h. Red colouration indicated NR fluorescence. (**b**) Quantification of NR fluorescence in zebrafish treated with PC70-NR or NR. * *p* < 0.05, ** *p* < 0.01. Each experimental group contained 12 fish; error bars indicate SD.

**Figure 3 molecules-25-01595-f003:**
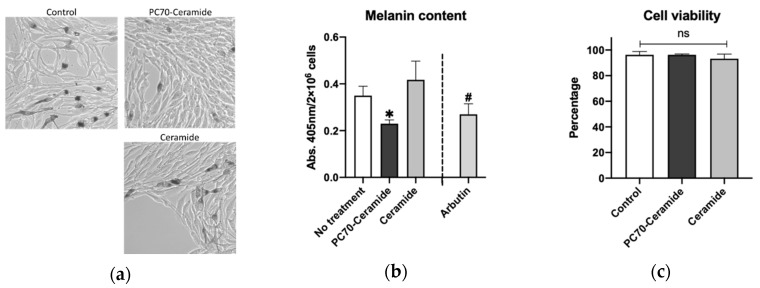
PC70-ceramide suppresses melanin accumulation in mouse B16 melanoma cells. (**a**) Representative images of B16 cells treated with PC70-ceramide or ceramide. B16 4A5 cells were treated with 0.5% PC70-ceramide (containing 50 µg/mL ceramide) or 50 μg/mL ceramide for 48 h. (**b**) Melanin levels of B16 cells treated with PC70-ceramide, ceramide, or arbutin. Arbutin was used as a positive control, following the method used by a previous study [8]. * *p* < 0.05, # *p* < 0.1 vs. control. n = 3, error bars indicate SD. (**c**) Cell viability of B16 cells treated with PC70-ceramide or ceramide. “ns” indicates no significance. n = 3, error bars indicate SD.

**Table 1 molecules-25-01595-t001:** LC_50_ of lecithin and lecithin-ceramides in zebrafish.

	Lecithin Alone	Lecithin-Ceramide
PC70	240.8 ± 34.6 µg/mL	533.1 ± 203.5 µg/mL (266.5 ± 101.8 µg/mL)
LPC70	3.0 ± 0.3 µg/mL	3.4 ± 0.4 µg/mL (1.7 ± 0.2 µg/mL)
LPC70H	1.1 ± 0.1 µg/mL	2.1 ± 1.0 µg/mL (1.1 ± 0.5 µg/mL)

The experiments were performed in triplicates independently, and each experimental group contained 12 fish. The values are shown as mean with SD. The values indicate lecithin concentrations in the emulsions and the numbers shown in ( ) indicate ceramide concentrations.

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
