# Peer review of "Lecithin-Based Dermal Drug Delivery for Anti-Pigmentation Maize Ceramide"

_molecules, 2020, doi:10.3390/molecules25071595_

Round 1
Reviewer 1 Report
This is an interesting report showing that maize ceramides can be delivered via lecithin-based emulsions. The authors have carefully designed and performed most of the necessary experiments and the results are appealing. Especially, the components of the emulsions are all very safe to the human. Therefore, I am happy to recommend the publication of the work in molecules if the authors can fully address the following minor issues:
- The role of the solvent glycerol in the dermal drug delivery should be mentioned. In addition, compared with DMSO, what are the advantages/disadvantages of glycerol in the dermal drug delivery?
- The optical microscopic or electron microscopic characterization of the formed emulsions should be conducted.
- The following related and recent references regarding the use of zebrafish as a model for evaluating the toxicity and drug delivery capability of materials should be cited in the Introduction: Chem. Sci. 2019, 10, 4062−4068; J. Control. Release 2019, 311−312, 301−318.
Reviewer 2 Report
A file with comments is attached. In short, there are several drawbacks in presenting the data and the data itself is incomplete.
The manuscript must be improved with detailed explanation for results section, and materials and methods section and based on this conclusions should be draw.

Round 2
Reviewer 2 Report
The authors have addressed the concerns and the articles is ready for publication.